# Social Media Usage among Dental Undergraduate Students—A Comparative Study

**DOI:** 10.3390/healthcare9111408

**Published:** 2021-10-20

**Authors:** Eswara Uma, Pentti Nieminen, Shani Ann Mani, Jacob John, Emilia Haapanen, Marja-Liisa Laitala, Olli-Pekka Lappalainen, Eby Varghase, Ankita Arora, Kanwardeep Kaur

**Affiliations:** 1Faculty of Dentistry, Manipal University College Malaysia, Melaka 75150, Malaysia; eswara.uma@manipal.edu.my (E.U.); eby.varghese@manipal.edu.my (E.V.); ankita.arora@manipal.edu.my (A.A.); kanwardeep.kaur@manipal.edu.my (K.K.); 2Medical Informatics and Data Analysis Research Group, University of Oulu, 90014 Oulu, Finland; 3Department of Paediatric Dentistry and Orthodontics, Faculty of Dentistry, University of Malaya, Kuala Lumpur 50603, Malaysia; shani@um.edu.my; 4Department of Restorative Dentistry, Faculty of Dentistry, University of Malaya, Kuala Lumpur 50603, Malaysia; drjacob@um.edu.my; 5Research Unit of Oral Health Sciences, Faculty of Medicine, University of Oulu, 90014 Oulu, Finland; emilia.haapanen@student.oulu.fi; 6Research Unit of Oral Health Sciences, Faculty of Medicine, University of Oulu and Medical Research Centre Oulu, Oulu University Hospital, 90014 Oulu, Finland; marja-liisa.laitala@oulu.fi; 7Faculty of Medicine, University of Helsinki, 00014 Helsinki, Finland; olli.pekka.lapplainen@helsinki.fi

**Keywords:** social media use, dental students, social media platforms, dental training, Malaysia, Finland

## Abstract

Social media use among students has infiltrated into dental education and offers benefits but may also cause problems. The aim of this study was to explore and compare current social media usage among dental undergraduate students from two countries—Malaysia and Finland. A self-administered structured online questionnaire was used. WhatsApp, YouTube, Instagram, Facebook and Snapchat were the services that were most familiar to the respondents from both countries. There were differences between the students from the two countries among the most preferred platforms. The most frequently used applications were WhatsApp (91.1% of students in Malaysia and 96.1% in Finland used it very frequently) and Instagram (74.3% of students in Malaysia and 70.0% in Finland used it very frequently). Students in Malaysia spent significantly more hours per week using the platforms as study tools than students in Finland. Over 80% of the Finnish dental students reported that lack of knowledge was not an issue in social media usage, while 85% of Malaysian students felt that lack of knowledge prevented them from using social media platforms frequently. The findings offer evidence that dental students used social media extensively.

## 1. Introduction

In today’s digital world, most people are logged in perpetually and always connected. Our devices have ensured that technology is “always on us and always on”! Never has it become so easy to access information, and social media has become a major tool for communication and seeking information. We have a multitude of interactions with others, on topics that can be varied, and this has fused our professional and personal lives [1].

Social media is defined as “websites and applications that enable users to create and share content or to participate in social networking.” Social media includes social networking platforms including Facebook and Twitter and media sharing sites, for example, YouTube and Instagram. In addition, there are other platforms like blog sites and micro-blogging sites. Healthcare professionals use social media extensively and it was reported that up to 90% of practicing doctors use Facebook accounts for professional or personal use [2]. Social media platforms offer different approaches to content sharing, and this has wide-ranging uses in dentistry. Social media is useful not only for education and networking, but also for marketing and recruitment. Studies in health professionals’ education have found benefits in the use of social media tools in clinical education [3]. Social media has become pervasive in society and is playing an important role in the personal and professional lives with dentistry being no exception.

In remote study conditions during the COVID-19 pandemic, the use of digital tools and social media platforms became imperative for medical and dental medical education for information retrieval, sharing of learning materials, and video meetings and discussions [4,5]. These tools were often used for both academic and non-academic purposes among students and teachers. Social media platforms were found to enhance ways of studying, allow for learning new skills, enhance performance, foster social relationships and social support and strengthen organizational identification [6,7]. 

Reports indicate that dental students use more than one social media application with Facebook being the most used platform among students in the United States and the United Kingdom [2,3,4,5,6,7,8]. The second most favored platforms were YouTube and Instagram. Skype and YouTube were used to improve dental skills while Twitter and blogging sites for interactions with the faculty and also to enhance communication [9].

A bibliometric study of articles published in journals indexed by the Web of Science database found 41 studies related to social media and dentistry during the period of 2010 –2016 [10]. Most of these studies focused on the impact of social media on dental education and professional practice. These studies emphasized the extension of the dental curriculum to involve the teaching and learning using social media platforms. However, these studies also highlighted the concern towards educational preparedness of future generations in the academic community, and understand the limitations of discourse produced by social media platforms. These studies also noted the concern regarding information mediation through social media platforms and its impact on dental education [10].

Significant differences in social trends, cultural beliefs and perceptions between Asian and European countries can affect the use of different social media applications and other technologies. This may also influence how these technologies are utilised by undergraduate students in their academics.

There are some previous studies about social media and dentistry in Malaysia [11,12,13,14]. Rani et al. [13] have described how dental undergraduates were trained to use social media for promoting oral health in the community, while Affendi et al. [11] evaluated the use of social media for marketing by dentists. See et al. [14] investigated the support, exposure and use of social media technologies among students, academics and administrators from both informatics and non-informatics undergraduate programs in Malaysia. In addition, internet addiction among dental students was also studied [12]. To our knowledge, there are no published studies about social media usage among Finnish dental or medical students.

The main purpose of our study was to compare the social media usage among dental undergraduate students from two countries Malaysia and Finland. This comparative study between similar cohorts of students from two different countries from two continents will help to estimate the extent and nature of social media use among undergraduate dental students. In addition, the study will provide suggestions for social media training in the dental curriculum. In this article, empirical data focused on the following research questions: How familiar are students with social media platforms? How often do students use social media services? How competent are the students at using social media services? How many hours do students spend using social media platforms as part of dental education? What factors encourage or prevent students to use social media?

## 2. Materials and Methods

### 2.1. Study Design and Data Collection

This was a cross-sectional online questionnaire survey conducted among dental undergraduates of the academic year 2020–2021 from two dental schools in Malaysia (Manipal Melaka Medical College and University of Malaya) and Finland (the University of Helsinki and University of Oulu) each. Ethical approval to conduct this study was obtained for the institutions in Malaysia; Medical Ethics Committee, Faculty of Dentistry, University of Malaya [DF CD2105/0015 (L)] and Research Ethics committee, Faculty of Dentistry, Melaka Manipal Medical college [MMMC/FOD/AR/E C-2021(F-01)] prior to commencement of the study. According to the guidelines of the Ministry of Education and Culture in Finland, survey studies with anonymous questionnaires do not need approval from an ethics committee.

The instrument used was a questionnaire modified from a previous study among dental students in the USA [15]. The validated questionnaire assesses social media usage among medical undergraduates. Twelve items from the questionnaire were used to assess social media usage and the perceptions of social media usage in relation to dentistry/dental practice. Questionnaire items had three sections; Part A consisted of five questions regarding the demographic characteristics of the participants. Part B had six items regarding their familiarity, competence, time spent on various social media platforms and factors that encouraged and discouraged students from using social media. Part C had one item which addressed practice and perceptions regarding social media use in dentistry. Most questions required responses on a 4-point or 5-point Likert scale. The updated version of the questionnaire is included as the Appendix A.

In Malaysia, the English questionnaire was pre-tested on a sample of five students in different years of the dental undergraduate programme at the Manipal Melaka Medical College and University of Malaya to check for semantic comprehension. Only minor modifications were made to the questionnaire following feedback from the pre-test. The questionnaire was translated into Finnish. In Finland, five dental students also pretested the first Finnish version. Based on their feedback, minor changes were made to improve the language and to clarify the purpose of the questions. The final questionnaire was administered using Google Forms from March 23rd, 2021, to April 11th, 2021, in both countries, the link being circulated via email and WhatsApp to student representatives of each year of study who then forwarded it to their classmates.

All dental undergraduates who received the online survey link were invited to participate in this study. In the online Google form, all the participants were asked to declare that they had read the participant information sheet (PIS) and voluntarily give consent for data collection and processing. If they refused consent, the questionnaire was closed. Inclusion and exclusion criteria were specified in the PIS. The survey was anonymous and did not include personal sensitive data.

For the estimation of sample sizes, we selected the time spent using social media as the outcome variable. The following formulas with finite population correction for proportions were used to estimate the minimum sample sizes in Malaysia and Finland: (1)$$n_{0}=\frac{z_{\alpha /2}^{2}p\left(1-p\right)}{e^{2}}$$
and
(2)$$n=\frac{n_{0}N}{n_{0}+\left(N-1\right)}$$
where *n* = required minimum sample size, *n*_0_ = Cochran’s sample size for large populations, *N* = available number of students (years 1–5) (population size), *e* = maximum error in estimation, *p* = proportion of the outcome variable (more than 15 h a week), *p*(1 − *p*) = variance of the outcome variable, *z*^2^*_α_*_/2_ = 1.96 for 95% confidence limit [16,17]. 

Setting maximum error to 5%, presuming that 50% of the student population have the outcome proportion of using more than 15 h a week social media and estimating that the population size *N* = 3250 in Malaysia, the minimum number of participating students should be at least 344 in Malaysia. Currently, there are about 1000 dental students (years 1–5) in Finland. So, the required sample size was 278 students in Finland.

### 2.2. Data Analysis

Tabular and graphical displays of data were used as the main tools of data presentation and analysis. The frequency and percentage distributions of participant characteristics (age, sex, year of study, and hours a week using social media) were presented for students from Malaysia and Finland. Percentage distributions were used to estimate the proportions of responses to questions “How familiar are you with each of the following social media services?”, “How often do you use each of the following social media services?”, “Approximately how many hours a week do you spend using the following social media services as part of your dental education?”, and to question “How competent are you with each of the following social media services?” by country. We also compared the percentage distributions of students using very frequently the five most popular social media platforms and how competent they felt using these platforms by age, sex and year of dental school. In addition, frequency and percentage distributions of factors that encouraged or prevented students to use social media platforms were presented. Statistical significance of differences between Malaysian and Finnish student groups and basic characteristics were evaluated using a chi-square test with exact *p*-values. Among applications, we also evaluated the relationship between overall high use, competence at using and use in dental education using Spearman’s rank correlation coefficient (rho). The data satisfactorily fulfilled the underlying assumptions and preconditions of the applied analysis methods. All statistical analyses were performed using IBM SPSS Statistics software (version 26) and Origin 2020 graphing software. 

## 3. Results

### 3.1. Participants

A total of 613 students participated in this study. Table 1 shows the distribution of age, sex, year of dental school and hours per week using social media by country. Most of the participants were female in both countries. The student groups were quite different in terms of age and year of dental school. In particular, the Malaysian students were younger than the Finnish dental students. Most of the students spent more than 11 h per week using social media. Almost 75% of the Finnish dental students reported that they used social media at least 11 h per week.

### 3.2. Familiarity with Social Media Platforms

Most of the students were familiar with several applications (Figure 1). WhatsApp, YouTube, Instagram, Facebook and Snapchat were the most familiar services to the respondents from both countries. All Malaysian and Finnish students were familiar or very familiar with WhatsApp. In addition, all Finnish students were at least familiar with YouTube and Facebook. The number of students not familiar with YouTube or Instagram was also minimal. Only two students from Malaysia reported that they were not familiar with YouTube. Seven Malaysian students and one Finnish student reported that they had heard about Instagram but were not sure of its purpose. Students in Malaysia were more aware of Telegram, WeChat and Weibo than students in Finland. Respectively, the Finnish students were more familiar with Facebook, Snapchat, Jodel and LinkedIn.

### 3.3. Reported Frequency of Social Media Use

Figure 2 shows that all students reported using more than one social media platform at least regularly. WhatsApp was the most commonly used, all students used it at least regularly, 91.1% very frequently in Malaysia and 96.1% in Finland. Instagram was the second most frequently used platform, 74.3% used it very frequently in Malaysia and 70.0% in Finland. Students in Malaysia were more likely to use YouTube, Telegram, Twitter, Google+ frequently or regularly than those in Finland. The Finnish students were more likely to use Snapchat and Jodel compared to students from than in Malaysia. Most of the respondents in our survey had never used WeChat, Tumblr, LinkedIn, or Weibo.

We also analysed the associations of background characteristics with the frequency of social media usage. For this analysis, we included only the five most popular platforms presented in Figure 2 and the analyses were stratified by country. Table 2 shows the proportions of respondents using frequently these basic student characteristics. The age of students was not associated with the frequent use of these services in Malaysia. However, younger students used Instagram and Snapchat more frequently than older students in Finland. 

In Malaysia, male participants were more likely to report very frequent use of Facebook than female participants (45.6% vs. 27.9%). In Finland, male students were more likely to report using very frequently YouTube than females (51.1% vs. 12.8%) and female students used Instagram (77.6% vs. 44.7%) or Snapchat (55.1% vs. 38.1%) more often than males. Year of study was not associated in either country with the very frequent use of the platforms (Table 2).

### 3.4. Perceived Competence of Social Media Use

Both Malaysian (99.6%) and Finnish (100.0%) students reported that they were highly competent or competent in using WhatsApp (Figure 3). We found some substantially significant differences in the reported competencies between students from Malaysia and Finland. More Malaysian students declared that they were highly competent in using YouTube, Instagram, Telegram, TikTok, Twitter and Google+. On the other hand, the Finnish respondents declared more often than they were highly competent using Facebook, Snapchat and Jodel. Figure 3 also shows that the majority of the students do feel that they are beginners or not at all competent in using Tumblr, LinkedIn or Weibo.

Table 3 shows the proportions of self-reported high competence in using the most preferred social media platforms by basic student characteristics and stratified by country. Sex was associated with the reported competence of using the most commonly used applications. Female students in both countries are more likely reported to be highly competent using popular social networking sites than males. In Malaysia, no substantial difference between males and females was found only in the self-reported skill of using Facebook. In Finland, males and females reported similar highly competent only in the use of YouTube. Year of study had little influence on the self-reported competence of using the most common platforms (Table 3). However, our analysis indicated that age was associated with the perceived high competence differently in Malaysia and Finland. In Malaysia, younger dental students felt less often highly competent with YouTube and Facebook. In Finland, younger students reported more likely that they were highly competent in applying Instagram and Snapchat. 

### 3.5. Reported Frequency of Social Media Use for Dental Education

Participants were also asked about the time (hours per week) they spent using social media sites as part of their dental education (Table 4). Frequency and percentage distributions by country show that there were statistically significant differences between the countries among the most preferred platforms. For most applications, students in Malaysia spent more hours per week using the platforms as study tools than students in Finland. Jodel was the only social media application where the Finns spent more hours per week to manage their study assignments or tutorials.

We also analysed whether students used the same social media platforms for their personal use as well as for their educational purposes and if their perceived competence correlated with the use of these specific platforms for dental education. Figure 4 shows a strong association between personal and educational use of platforms in both countries. In addition, perceived knowledge of the use of the applications was associated with their use in educational purposes.

### 3.6. Encouraging or Preventing Factors to Use Social Media 

Table 5 reports factors that encouraged students to use social media platforms. To stay in touch with friends and family members was very important in both countries (77.0% in Malaysia vs. 86.2% in Finland). However, Malaysian students valued platforms more in connecting with old friends they had lost touch with than Finnish students. Malaysian students found social media platforms also more encouraging in communicating about issues related to dental training (Table 5).

We also asked students about factors that prevented them from using social media. Students from Malaysia reported more often reasons that they experienced important not to use social media than students from Finland (Figure 4). Over 80% of the Finnish dental students reported that lack of knowledge was not an issue in social media usage, while 85% of Malaysian students felt that lack of knowledge prevented them from using social media platforms somewhat or very much. Similar differences were observed with lack of time, lack of interest, lack of perceived value and concern about harm to professional image (Figure 5).

## 4. Discussion

The present comparative study between Malaysian and Finnish universities was conducted to evaluate the social media usage among dental undergraduate students. We found that the same top five platforms (WhatsApp, YouTube, Instagram, Facebook and Snapchat) were the most familiar services to the respondents from both countries. There were country-specific differences among students in the familiarity and usage of the other platforms. For most applications, students in Malaysia spent more hours per week using the platforms as study tools than students in Finland. Malaysian students found social media platforms more encouraging in communicating about issues related to dental training. Finnish dental students reported more often that lack of knowledge was not an issue in social media usage, while the majority of Malaysian students felt that lack of knowledge prevented them from using social media platforms somewhat or very much. Age and year of dental studies were not clearly associated in either country with the frequent use of the platforms.

Most students from both countries used social media for more than eleven hours per week. In addition, 25.3% of the respondents used social media more than twenty hours per week. This is not surprising but reflects the current habits of communication among young adults. Recent studies from different fields of study have reported that undergraduate students spent on average about three to four hours per day using social media platforms [18,19,20,21]. These services include mainly entertainment and communication, but also learning and searching for general information. 

Our findings reflect the global popularity of the same platforms in contemporary society and show that most students are now heavy users of these platforms in the two culturally different countries. Most of the participants from both countries were familiar with WhatsApp, YouTube, Instagram, and Facebook. In a global world, students are also aware of other several social media services, but do not necessarily use all of them. There was a difference in the familiarity with some of the social media applications by the students in both countries. Malaysian students were more familiar with Telegram, WeChat, and Weibo while Finnish students were more familiar with Snapchat, Jodel and LinkedIn. This difference could be attributed to the students being familiar with the social media applications that are more commonly used in their region. Probably most of the students use those applications that their peers use so that there is an ease in communication and sharing. Jodel is a community-based social media application and is limited to its use in Europe and part of the Middle East [22]. LinkedIn is a social platform that mainly caters to people involved in business and has so far been very popular with the older age group [23] and among dental professionals looking to develop connections [24]. Finnish students being slightly older could account for their greater familiarity with LinkedIn. Familiarity with Telegram and its usage has grown exponentially in Malaysia as it is a popular medium to disseminate official information from Government [25]. There was no statistically significant difference in the usage of social media based on age, sex, and years of study in both the groups of students, though female students showed a greater tendency to use Instagram than the male which is similar to the worldwide data [26].

When considering the prevalence of using social media services among dental students, it is worth noting that these services have different purposes and target audiences. For example, the students in our sample may be too old to use Snapchat, a popular chat and communication channel for children and young people, for daily communication. On the other hand, they are still too young and without sufficient work experience to seek new work contacts through LinkedIn. The popularity of WhatsApp among study participants reflects the importance of keeping in touch with friends, family members or members of other restricted messaging groups in a convenient way (whether by text, picture or voice). Instagram and Twitter offer channels to increase internet online visibility. Using YouTube involves sharing your own videos and watching and commenting on other users’ videos. 

It was observed that all the students had greater competency in the social media applications they used most frequently which was primarily WhatsApp. Between both the groups of students, the difference observed regarding competence in using social media was the same as observed related to social media usage. Application usage by individual increases as he/she becomes more familiar with its features and thereby feels competent in using the application. Age of the student, year of study did not show any significance regarding the perception of competence for using social media applications, though females showed statistically significant competence in using Instagram, which seems to be very popular among females [26].

Our research clearly showed that students prefer to use for educational purposes the same platforms as they use for personal communication and online visibility. An interesting finding was spending a greater number of hours per week on the most popular social media applications like WhatsApp, YouTube, Instagram, and Facebook by the Malaysian students for purposes related to their dental education. This was statistically significant compared to Finnish students who used more hours on Jodel for the assignments or tutorials. This contrasts with a study from Saudi where dental students preferred to use Facebook for their learning [21]. 

The literature is not conclusive about the effects of social media on healthcare education. Some studies have reported a positive impact of social media on education while some have mentioned the reluctance of students to use social media for their education due to perceived negative impacts [27,28,29,30]. There is a study on the use of social media for dental health promotion [13], for dental marketing [11] and for dental education [31], as well as a review on what has been carried out so far in literature [10]. Literature also has studies on the usage of social media by dental students and its effects [21,29,32], however, in Malaysia and Finland, it is not known how dental students are using social media. It is known that the dental students of the present cohort are digital natives, they use social media universally and their dental education includes information retrieval studies using online platforms [33]. In Finland, all study programs in the participating faculties require the use of social media in different forms now. Social media is an integral part of modern pedagogy along with other methods. The dental schools of this study in Malaysia (Manipal Melaka Medical College and the University of Malaya) and Finland (the University of Helsinki and the University of Oulu) follow the official guidelines for social media use and learning these is also an elementary study content of every dental student.

Dental students from both countries considered keeping in touch with family and friends as a very important factor that encouraged them to use social media. This is probably why WhatsApp as a communication tool is so popular. It should be noted that the effect of social media tools may affect soft skills such as communication and confidence in spoken language in the future among these students. 

For Malaysian students, the reasons for refraining from the use of social media applications such as lack of knowledge about a particular application of interest, lack of time, lack of interest, lack of perceived value and concern about harmful effects on professional image, were more important than to Finnish students. In Malaysia, as per the MCMC report, one of the top reasons for Malaysians not using the internet in 2020 was a lack of interest [25]. The same report also mentioned that while the majority of Malaysians’ online activity is sending text messages to communicate and visit social media sites, sharing of content using social media has gone up since 2018. 

The extensive use of social media may have several drawbacks. Dental students should be made aware of the quality of information that they obtain from social media [33,34]. Social media content related to education may not be evidence-based or referenced from reliable sources since anyone can upload content that can affect not only their learning but also the profession [21]. Using social media can be time-consuming, addictive and distract from studying [19,20,21]. The social media activity of some healthcare students has also resulted in unanticipated ethical consequences [6]. In our study, questions were not specific to dental education and most of the questions were regarding the general use of social media. So, it would be inappropriate to extrapolate the questions of our study to social media use in dental learning. We will explore the disadvantages of using social media, long-term consequences, and perceptions of e-professionalism among dental students in our further study.

The present study used only a self-completed questionnaire and may suffer from the disadvantages of a cross-sectional online survey. The questionnaire failed to explain the underlying reasons for using specific platforms. In addition, the cross-sectional nature of our study made it impossible to assess possible rapid changes in students’ use and preferences of social media. We did not study how social media communication has affected study engagement before and during the COVID-pandemic. Thus, we do not know whether and how the pandemic changed students’ social media use. Future studies will need to study trends and priorities of social media use among dental students. Another limitation of this study should be noted. It was performed in local settings in Malaysia and Finland. Each regional setting is unique. Despite its limited scope, our findings might be helpful in considering the use of social media in different forms, and especially when considering students’ attitudes, preferences, and experiences with various platforms.

## 5. Conclusions

This multi-institutional study provides useful information on the usage of social media during the COVID-pandemic among dental students in two culturally different countries. The findings offer evidence that dental students used social media extensively in both countries. A few apps, which are popular worldwide, are widely used by students for both personal communication and education. Regionally popular platforms bring variety to the social media toolbox of dental students. Extensive use of social media can also be a distraction, especially when used for non-educational purposes. Students should be guided if they have specific interests or lack knowledge in some respects.

## Figures and Tables

**Figure 1 healthcare-09-01408-f001:**
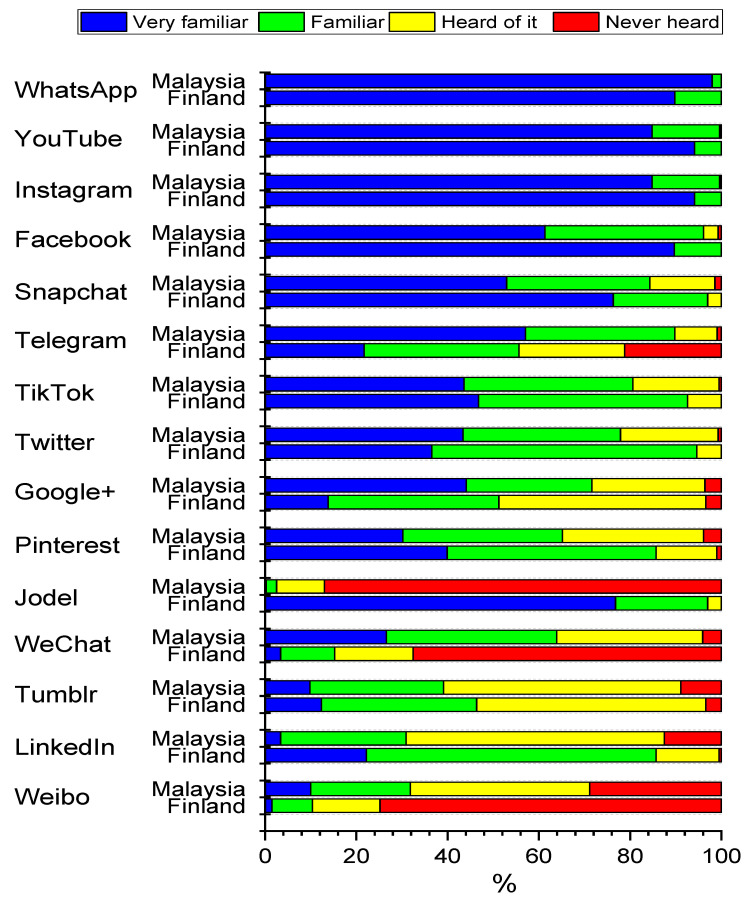
Percentage distributions of responses to question “How familiar are you with each of the following social media services?” by country. Data include dental undergraduate students from Malaysia (*n* = 440) and Finland (*n* = 203). Statistical significances between countries evaluated by exact chi-square test are as follows: 0.059 (Instagram), 0.054 (Tumblr), 0.002 (YouTube), <0.001 (all other services).

**Figure 2 healthcare-09-01408-f002:**
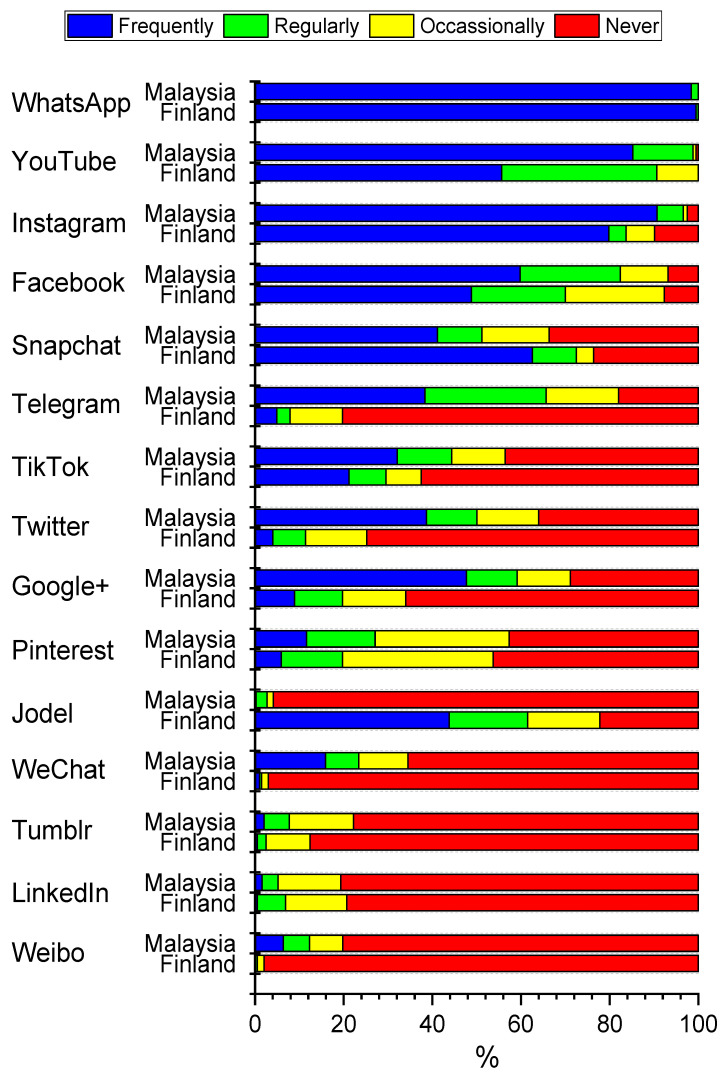
Percentage distributions of responses to question “How often do you use each of the following social media services?” by country. Data include dental undergraduate students from Malaysia (*n* = 440) and Finland (*n* = 203). Statistical significances between countries evaluated by exact chi-square test are as follows: 0.304 (LinkedIn), 0.294 (WhatsApp), 0.118 (Pinterest), 0.014 (Tumblr), 0.002 (Facebook), <0.001 (all other services).

**Figure 3 healthcare-09-01408-f003:**
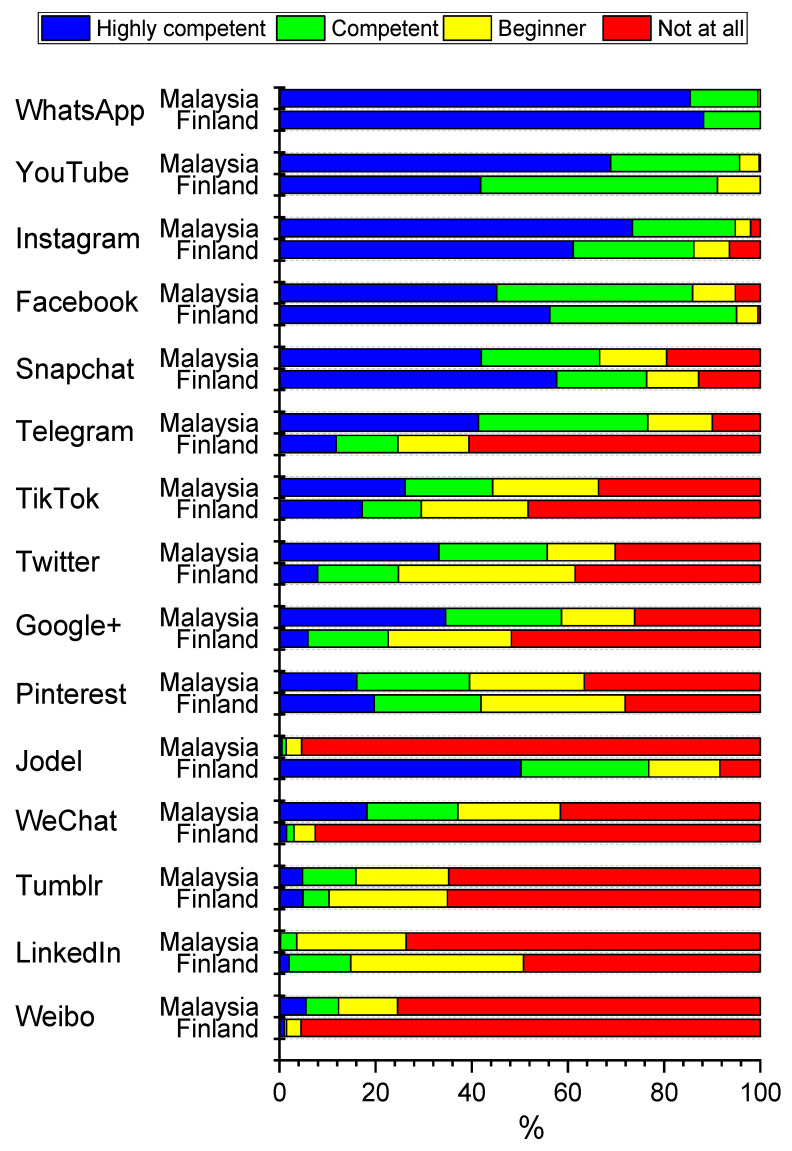
Percentage distributions of responses to question “How competent are you with each of the following social media services?” by country. Data include dental undergraduate students from Malaysia (*n* = 440) and Finland (*n* = 203). Statistical significances between countries evaluated by exact chi-square test are as follows: 0.426 (WhatsApp), 0.105 (Pinterest), 0.080 (Tumblr), 0.003 (Snapchat), <0.001 (all other services).

**Figure 4 healthcare-09-01408-f004:**
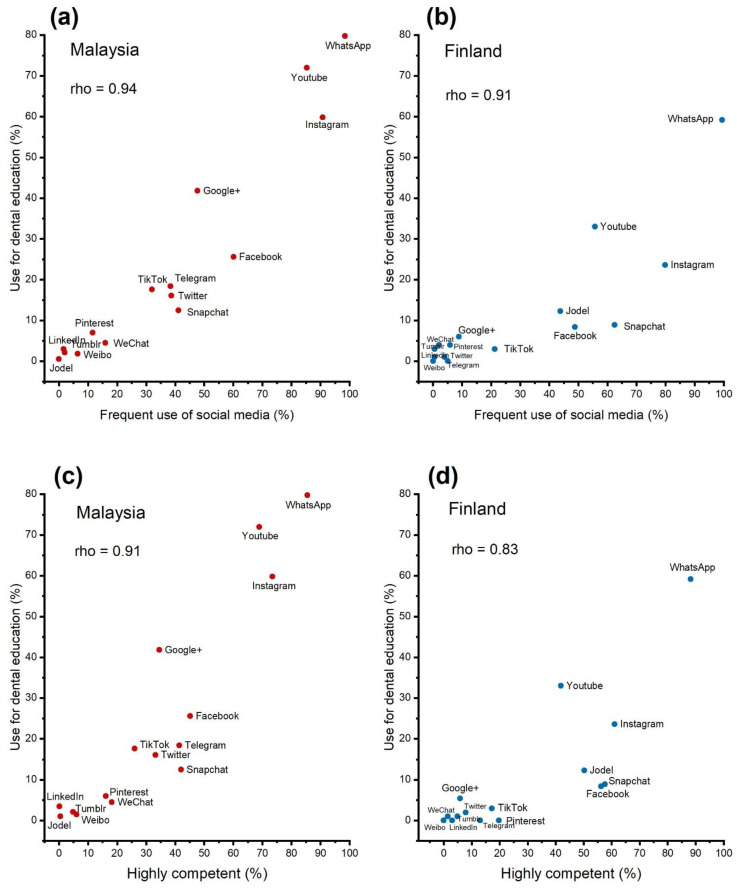
(**a**) Scatter plot showing correlation between percentage frequency of use of social media platforms (frequent and very frequent) for personal use vs. use in dental education in Malaysia and (**b**) in Finland; (**c**) Scatter plot showing correlations between perceived competence (highly competent) of use of social media platforms vs. use in dental education in Malaysia, and (**d**) in Finland. Data include dental undergraduate students from Malaysia (*n* = 440) and Finland (*n* = 203). rho = Spearman’s rank correlation coefficient.

**Figure 5 healthcare-09-01408-f005:**
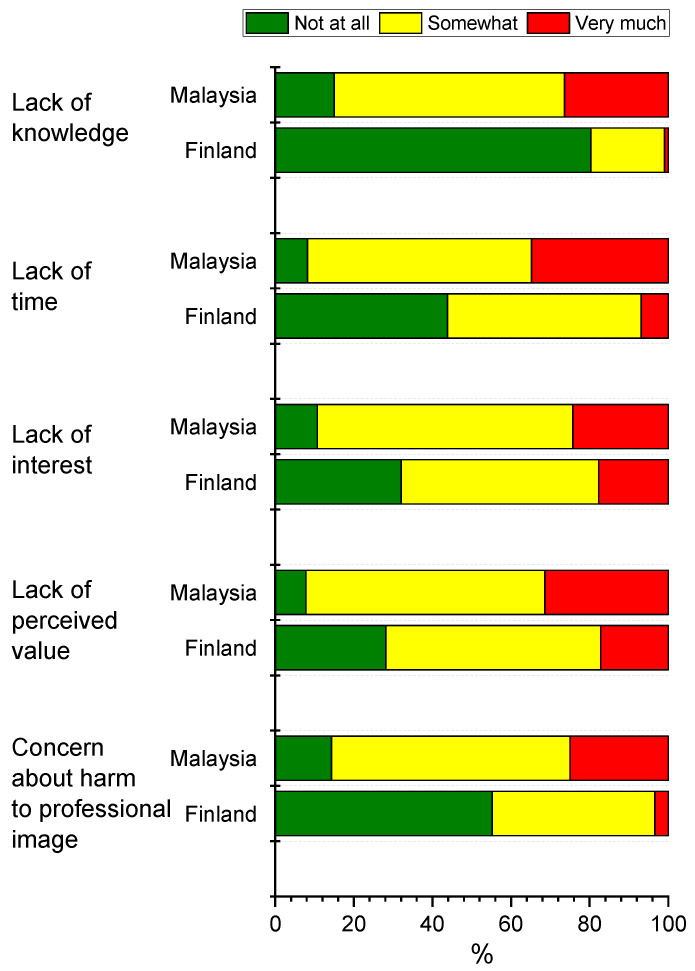
Percentage distributions of responses to question “How much the following reasons (factors) prevent you from using social media?” by country. Data include dental undergraduate students from Malaysia (*n* = 440) and Finland (*n* = 203). There were statistically significant differences (*p* < 0.001) between the countries among all reasons (factors). Significances were evaluated by chi-square test.

**Table 1 healthcare-09-01408-t001:** The frequency and percentage distributions of basic characteristics among dental students from Malaysia (*n* = 440) and Finland (*n* = 203).

	Country			*p*-Value of Exact Chi-Square Test
	Malaysia	Finland	All	
Characteristics	*n* (%)	*n* (%)	*n* (%)	
Age				<0.001
20 years or younger	70 (17.0)	12 (5.9)	87 (13.5)	
21–23 years	250 (56.8)	56 (27.6)	306 (47.6)	
24–26 years	114 (25.9)	69 (34.0)	183 (28.5)	
27–29 years	1 (0.2)	38 (18.7)	39 (6.1)	
30 years or above	0	28 (13.8)	28 (4.4)	
Sex				>0.999
Male	103 (23.4)	47 (23.2)	150 (23.3)	
Female	337 (76.6)	156 (76.7)	493 (76.7)	
Year of dental school				0.038
First	70 (15.9)	39 (19.2)	109 (17.0)	
Second	78 (17.7)	54 (26.6)	132 (20.5)	
Third	79 (18.0)	33 (16.3)	112 (17.4)	
Fourth	108 (24.5)	42 (20.7)	150 (23.3)	
Fifth	105 (23.9)	35 (17.2)	140 (21.8)	
Hours a week using social media				<0.001
Do not use	1 (0.2)	2 (1.0)	3 (0.5)	
Less than 5 h	51 (11.6)	7 (3.4)	58 (9.0)	
6–10 h	114 (25.9)	42 (20.7)	156 (24.3)	
11–15 h	81 (18.4)	62 (30.5)	143 (22.2)	
16–20 h	76 (17.3)	44 (21.7)	120 (18.7)	
More than 20	117 (26.6)	46 (22.7)	163 (25.3)	

**Table 2 healthcare-09-01408-t002:** Percentage of students using very frequently the five most popular social media platforms by age, sex and year of dental school. Data include dental undergraduate students from Malaysia (*n* = 440) and Finland (*n* = 203).

	Platform					
	WhatsApp	YouTube	Instagram	Facebook	Snapchat	Number of Students
Characteristics	%	%	%	%	%	
Malaysia						
Age	*p* = 0.725	*p* = 0.556	*p* = 0.778	*p* = 0.333	*p* = 0.185	
20 years or younger	90.7	66.7	77.3	26.7	33.3	75
21–23 years	90.4	59.6	72.8	32.4	26.0	250
24–26 years	93.0	57.9	75.4	34.2	24.6	114
27–29 years	100.0	100.0	100.0	100.0	100.0	1
30 years or above	-	-	-	-	-	0
Sex	*p* = 0.073	*p* = 0.135	*p* = 0.369	*p* = 0.001	*p* > 0.999	
Male	86.4	67.0	70.9	45.6	27.2	103
Female	92.6	58.5	75.4	27.9	27.0	337
Year of dental school	*p* = 0.462	*p* = 0.109	*p* = 0.304	*p* = 0.629	*p* = 0.524	
First	92.9	65.7	77.1	27.1	30.0	70
Second	92.3	70.5	76.9	37.2	21.8	78
Third	88.6	59.5	64.6	30.4	22.8	79
Fourth	88.0	51.9	75.9	29.6	31.5	108
Fifth	94.3	59.0	76.2	35.2	27.6	105
Finland						
Age	*p* = 0.184	*p* = 0.214	*p* = 0.025	*p* = 0.860	*p* < 0.001	
20 years or younger	83.3	16.7	83.3	16.7	75.0	12
21–23 years	98.2	14.3	78.6	32.1	82.1	56
24–26 years	97.1	24.6	63.8	33.3	49.3	69
27–29 years	94.7	18.4	78.9	31.6	39.5	38
30 years or above	96.4	35.7	50.0	32.1	0.0	28
Sex	*p* > 0.999	*p* < 0.001	*p* < 0.001	*p* = 0.107	*p* = 0.047	
Male	95.7	51.1	44.7	21.3	38.3	47
Female	96.2	12.8	77.6	34.6	55.1	153
Year of dental school	*p* = 0.131	*p* = 0.285	*p* = 0.509	*p* = 0.400	*p* = 0.155	
First	89.7	23.1	66.7	23.1	56.4	39
Second	98.1	14.8	66.7	27.8	61.1	54
Third	93.9	18.2	75.8	30.3	48.5	33
Fourth	100.0	21.4	78.6	35.7	50.0	42
Fifth	97.1	34.3	62.9	42.9	34.3	35

Statistical significances evaluated by exact chi-square test for each platform.

**Table 3 healthcare-09-01408-t003:** Percentage of students reporting high competence in using the five most commonly used social media platforms by age, sex and year of dental school. Data include dental undergraduate students from Malaysia (*n* = 440) and Finland (*n* = 203).

	Platform					
	WhatsApp	YouTube	Instagram	Facebook	Snapchat	Number of Students
Characteristics	%	%	%	%	%	
Malaysia						
Age	*p* = 0.774	*p* = 0.040	*p* = 0.537	*p* = 0.009	*p* = 0.294	
20 years or younger	88.0	78.7	78.7	32.0	49.3	75
21–23 years	84.4	69.6	73.6	44.8	40.8	250
24–26 years	86.0	60.5	69.3	54.4	39.5	114
27–29 years	100.0	100.0	100.0	100.0	100.0	1
30–35 years	-	-	-	-	-	0
Sex	*p* = 0.001	*p* = 0.010	*p* < 0.001	*p* > 0.999	*p* = 0.255	
Male	74.8	58.3	56.3	45.6	36.9	103
Female	88.7	72.1	78.6	45.1	43.6	337
Year of dental school	*p* = 0.347	*p* = 0.013	*p* = 0.313	*p* = 0.142	*p* = 0.430	
First	88.6	81.4	80.0	32.9	45.7	70
Second	88.5	78.2	79.5	46.2	44.9	78
Third	84.8	64.6	68.4	41.8	34.2	79
Fourth	79.6	63.9	71.3	50.0	46.3	108
Fifth	87.6	61.9	70.5	50.5	39.0	105
Finland						
Age	*p* = 0.015	*p* = 0.079	*p* = 0.009	*p* = 0.322	*p* < 0.001	
20 years or younger	75.0	50.0	66.7	41.7	66.7	12
21–23 years	91.1	44.6	71.4	55.4	78.6	56
24–26 years	94.2	50.7	65.2	63.8	66.7	69
27–29 years	89.5	23.7	57.9	57.9	42.1	38
30–35 years	71.4	35.7	32.1	42.9	10.7	28
Sex	*p* = 0.009	*p* = 0.867	*p* < 0.001	*p* = 0.001	*p* = 0.019	
Male	76.6	40.4	31.9	34.0	42.6	47
Female	91.7	42.3	69.9	62.8	62.2	156
Year of dental school	*p* = 0.568	*p* = 0.659	*p* = 0.339	*p* = 0.529	*p* = 0.311	
First	87.6	38.5	61.5	43.6	56.4	39
Second	87.0	44.4	61.1	59.3	59.3	54
Third	97.0	51.5	75.8	60.6	72.7	33
Fourth	85.7	40.5	52.4	57.1	52.4	42
Fifth	85.7	34.3	57.1	60.0	48.6	35

Statistical significances evaluated by exact chi-square test for each platform.

**Table 4 healthcare-09-01408-t004:** Frequency and percentage distributions of responses to question “Approximately how many hours a week do you spend using the following social media services as part of your dental education?” by country. Data include dental undergraduate students from Malaysia (*n* = 440) and Finland (*n* = 203).

Platform	Country	Do Not Use *n* (%)	Less than 1 h *n* (%)	1–5 h *n* (%)	6–10 h *n* (%)	More than 11 h *n* (%)	*p*-Value of Exact Chi-Square test
WhatsApp	Malaysia Finland	26 (5.9) 6 (3.0)	63 (14.3) 77 (37.9)	155 (35.2) 101 (49.8)	91 (20.7) 14 (6.9)	105 (23.9) 5 (2.5)	<0.001
YouTube	Malaysia Finland	19 (4.3) 35 (17.2)	104 (23.6) 101 (49.8)	204 (46.4) 53 (26.1)	71 (16.1) 10 (4.9)	42 (9.5) 4 (2.0)	<0.001
Instagram	Malaysia Finland	31 (7.0) 77 (37.9)	146 (33.2) 78 (38.4)	157 (35.7) 36 (17.7)	50 (11.4) 7 (3.4)	56 (12.7) 5 (2.5)	<0.001
Facebook	Malaysia Finland	187 (42.5) 62 (30.5)	140 (31.8) 124 (61.1)	75 (17.0) 16 (7.9)	22 (5.0) 1 (0.5)	16 (3.6) 0	<0.001
Snapchat	Malaysia Finland	334(75.9) 141 (69.5)	51 (11.6) 44 (21.7)	34 (7.7) 15 (7.4)	7 (1.6) 1 (0.5)	14 (3.2) 2 (1.0)	0.006
Telegram	Malaysia Finland	260 (59.1) 187 (92.1)	99 (22.5) 16 (7.9)	42 (9.5) 0	21 (4.8) 0	18 (4.1) 0	<0.001
TikTok	Malaysia Finland	304 (69.1) 173 (85.2)	58 (13.2) 24 (11.8)	38 (8.6) 3 (1.5)	18 (4.1) 2 (1.0)	22 (5.0) 1 (0.5)	<0.001
Twitter	Malaysia Finland	281 (63.9) 183 (90.1)	88 (20.0) 19 (9.4)	48 (10.9) 1 (0.5)	9 (2.0) 0	14 (3.2) 0	<0.001
Google+	Malaysia Finland	197 (44.8) 172 (84.7)	59 (13.4) 20 (9.9)	91 (20.7) 10 (4.9)	49 (11.1) 1 (0.5)	44 (10.0) 0	<0.001
Pinterest	Malaysia Finland	349 (79.3) 175 (86.2)	65 (14.8) 29 (13.8)	22 (5.0) 0	2 (0.5) 0	2 (0.5) 0	0.007
Jodel	Malaysia Finland	435 (98.9) 112 (55.2)	1 (0.2) 66 (32.5)	2 (0.5 23 (11.3))	0 2 (1.0)	2 (0.5) 0	<0.001
WeChat	Malaysia Finland	402 (91.4) 192 (94.6)	18 (4.1) 11 (5.4)	12 (2.7) 0	5 (1.1) 0	3 (0.7) 0	0.036
Tumblr	Malaysia Finland	414 (94.1) 192 (94.6)	17 (3.9) 11 (5.4)	6 (1.4) 0	1 (0.2) 0	2 (0.5) 0	0.275
LinkedIn	Malaysia Finland	409 (93.0) 176 (86.7)	19 (4.3) 27 (13.3)	11 (2.5) 0	0 0	1 (0.2) 0	<0.001
Weibo	Malaysia Finland	419 (95.2) 185 (91.1)	13 (3.0) 18 (8.9)	6 (1.4) 0	1 (0.2) 0	1 (0.2) 0	0.003

**Table 5 healthcare-09-01408-t005:** Frequency and percentage distributions of responses to questions “How important is each of the following factors in encouraging you to use social media?” by country. Data include dental undergraduate students from Malaysia (*n* = 440) and Finland (*n* = 203).

Factor	Country	Not at All Important *n* (%)	Somewhat Important *n* (%)	Very Important *n* (%)	I Don’t Use Social Media *n* (%)	*p*-Value of Exact Chi-Square Test
To stay in touch with current friends and family members	Malaysia Finland	5 (1.1) 5 (2.5)	91 (20.7) 23 (11.3)	339 (77.0) 175 (86.2)	5 (1.1) 0	0.006
To connect with old friends, I have lost touch with	Malaysia Finland	20 (4.5) 44 (21.7)	195 (44.3) 113 (55.7)	219 (49.8) 42 (20.7)	6 (1.4) 4 (2.0)	<0.001
To connect around a shared hobby	Malaysia Finland	61 (13.9) 33 (16.3)	200 (45.5) 89 (43.8)	166 (37.7) 75 (36.9)	13 (3.0) 6 (3.0)	0.884
To communicate about issues relating to dental training	Malaysia Finland	13 (3.0) 27 (13.3)	198 (45.0) 97 (47.8)	217 (49.3) 79 (38.9)	12 (2.7) 0	<0.001

## Data Availability

The data presented in this study are available on request from the corresponding author.

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
