# Peer review of "Social Media Usage among Dental Undergraduate Students—A Comparative Study"

_healthcare, 2021, doi:10.3390/healthcare9111408_

Round 1

Reviewer 1 Report

This is a well-researched, nicely written and strongly argued article. 

  1. Please explain the sample size formula more fully on page 3 lines 144-147. What does no  represent - please advise.
  2. On page 14 line 333, remove the space after "[24],".
  3. On page 16 line 439, remove "to" after "should". 

Author Response

This is a well-researched, nicely written and strongly argued article.

Thank you.

Please explain the sample size formula more fully on page 3 lines 144-147. What does no  represent - please advise.

Sample size n0 was developed by Cochrane (ref 17) for large (or infinite) populations. If the population is finite, then the sample size can be reduced slightly. The sample size (n0) can be adjusted using equation for n as we have done. We have now clarified the sample size equations in the study design and data collection section.

On page 14 line 333, remove the space after "[24],"..

We have corrected the typo.

On page 16 line 439, remove "to" after "should”.

We have corrected this misspelling.

Reviewer 2 Report

First of all, I would like to thank the authors for the effort they have made in researching this topic. It is a very interesting study presenting social media usage among dental undergraduate students.

However, I would like to recommend some considerations: 

Abstract: - too long, the maximum length is 200 words.

  • Include some statistical values in results section of the abstract
  • Conclusion: „Social media is a useful tool for educational purposes, however students need to be made aware of the quality of information that they obtain from social media” – does not refer specifically to the results of the current study results. – please rephrase.

Introduction: - correctly presented

Materials and methods:- correct

Results:- an error in Table 1: first row: 20 years or younger- for “all” the value is 87 and does not correspond to the sum of Finnish and Malaysian participants (12+70).

 Discussions

  • Present some disadvantages of using social media and long-term consequences
  • Include some limitations of the study (eg. Including supplementary questions).

Conclusions- it is necessary to rephrase of the entire paragraph. Please limit the conclusions to the results of the current study and do not present any general conclusions. I consider the statement “In future, textbooks may become irrelevant” is inappropriate and hazardous.

-“ We conclude that visual aids such as YouTube and Instagram make learning easy, rather than reading textual material” does not refer to the results of the current study. – such question was not included in the questionary.

Author Response

First of all, I would like to thank the authors for the effort they have made in researching this topic. It is a very interesting study presenting social media usage among dental undergraduate students.

Thank you.

Abstract: - too long, the maximum length is 200 words.

We have now shortened the abstract.

Include some statistical values in results section of the abstract:

We have added the percentages of very frequent use of WhatsApp and Instagram in the abstract.

Conclusion: „Social media is a useful tool for educational purposes, however students need to be made aware of the quality of information that they obtain from social media” – does not refer specifically to the results of the current study results. – please rephrase.

We agree and we have deleted this sentence.

Introduction: - correctly presented

Thank you.

Materials and methods:- correct.

Thank you.

Discussions: Present some disadvantages of using social media and long-term consequences. Include some limitations of the study (eg. Including supplementary questions).

To present some disadvantages of using social media we have added the following text in the discussion:

“The extensive use of social media may have several drawbacks. Dental students should be made aware of the quality of information that they obtain from social media [33,34]. Social media content related to education may not be evidence-based or referenced from reliable sources since anyone can upload content that can affect not only their learning but also the profession [21]. Using social media can be time consuming, addictive and distract from studying [19–21]. The social media activity of some healthcare students has also resulted in unanticipated ethical consequences [6]. In our study, questions were not specific to dental education and most of the questions were regarding general use of social media. So, it would be inappropriate to extrapolate the questions of our study to social media use in dental learning. We will explore disadvantages of using social media, long-term consequences, and perceptions of e-professionalism among dental students in our further study.”

We have included the following limitations in the discussion:

“The present study used only self-completed questionnaire and may suffer from the disadvantages of a cross-sectional online survey. The questionnaire failed to explain the underlying reasons for using specific platforms. In addition, the cross-sectional nature of our study made it impossible to assess possible rapid changes in students' use and preferences of social media. We did not study how social media communication has affected study engagement before and during the COVID-pandemic. Thus, we do not know whether and how the pandemic changed students’ social media use. Future studies will need to study trends and priorities of social media use among dental students. Another limitation of this study should be noted. It was performed in local settings in Malaysia and Finland. Each regional setting is unique. Despite its limited scope, our findings might be helpful in considering use of social media in different forms, and especially when considering students' attitudes, preferences, and experiences with various platforms.”

Conclusions: - it is necessary to rephrase of the entire paragraph. Please limit the conclusions to the results of the current study and do not present any general conclusions. I consider the statement “In future, textbooks may become irrelevant” is inappropriate and hazardous.

-“ We conclude that visual aids such as YouTube and Instagram make learning easy, rather than reading textual material” does not refer to the results of the current study. – such question was not included in the questionary.

We have revised this text to read:

“This multi-institutional study provides useful information on the usage of social media during the COVID-pandemic among dental students in two culturally different countries. The findings offer evidence that dental students used social media extensively in both countries. A few apps, which are popular worldwide, are widely used by students for both personal communication and education. Regionally popular platforms bring variety to the social media toolbox of dental students. Extensive use of social media can also be a distraction, especially when used for non-educational purposes. Students should be guided if they have specific interests or lack of knowledge in some respects.”

Reviewer 3 Report

The manuscript by Uma et al addresses an important and interesting topic, the use of social media for medical and dental education. Unfortunately, in its present form it only provides a minuscule advance and delivers very few new insights.

On the positive side, I have to laude the authors for comparing the use of social media between two different groups of students, one from a developed country and the other from a developing country, each represented by two different schools/universities. However, disappointing is that the study concentrates on social media use in general and barely touches upon social media use for dental educational, being represented by only one question in the surveys.

The authors present a lot of data, but address/answer few meaningful questions and it remains open what their findings mean for dental education. The use of social media by dental students is not new and not surprising. Also, the observed differences between Finland and Malaysia could be more regional and be related to different social media platforms used and being popular in different global regions. Unless these two countries represent larger areas, very few people will care about country-specific findings or a comparison between two specific countries.

The discussion reads more like a general review and not like a careful discussion of the findings. A discussion should also not be a reiteration of the findings, but rather address what the findings mean in the context of the wider field (dental education) and compare them to the published findings of other researchers. In summary, the discussion gets lost in small details and lacks a wider vision.

Even though the surveys were conducted during the COVID pandemic, there is no comparison to pre-pandemic social media use. Therefore, we really do not know whether and how the pandemic changed students’ social media use. Shorten all references to the COVID pandemic.

There are a few interesting questions the authors should be able to address with their dataset. For example, are the social media platforms students use for their personal use correlated or the same as the platforms they use for educational purposes? Also, is their feeling of competence correlated to the use of specific platforms for dental education? Answering these questions would elevate the manuscript considerably. Also consider carefully, what of your data are really of interest to potential readers. Do not overload the manuscript with a lot of numbers, most having very little relevance. Less is often more. It might help to formulate a clear working hypothesis at the end of the introduction. What did you set out to learn, which questions did you want to answer with your research? The current purpose of the study would not encourage me to read the entire paper.

Some smaller points:

Please clarify whether social media were officially used during any courses at any of the four schools sampled in this study.

Be careful when stating the “social media are useful for education”. The evidence is contradictory as is later mentioned in your manuscript. Also, you do NOT investigate the efficacy of social media in your project. It would be nice to mention that social media can also be a distraction, especially when used for non-educational purposes.

A paragraph outlining the limitations of the study should be added at the end of the discussion, preceding the conclusions.

It would be helpful to add effect sizes to the statistical analyses.

Chi-square test is not the most appropriate statistical test for small response sizes. A Fisher’s exact test is recommended for such outcomes. Please consult a statistics expert.

Author Response

We are grateful to reviewer #3 for the close reading of the article.

The manuscript by Uma et al addresses an important and interesting topic, the use of social media for medical and dental education. Unfortunately, in its present form it only provides a minuscule advance and delivers very few new insights.

We appreciate this comment. We have now clarified the main aims and findings of our study.  

On the positive side, I have to laude the authors for comparing the use of social media between two different groups of students, one from a developed country and the other from a developing country, each represented by two different schools/universities. However, disappointing is that the study concentrates on social media use in general and barely touches upon social media use for dental educational, being represented by only one question in the surveys.

Our questionnaire included three parts: 1. variables related to the basic background factors of the participants, 2. Questions related to the social media usage in general, and 3. questions related to e-professionalism and use of social media in dental education. We cannot report all the results in one article, it would be too long. We are currently working on a second article, focusing on perceptions of e-professionalism and on the challenges that dental students may face due to the use of communication technologies in dental education. This is now described in the discussion as follows:

“In our study, questions were not specific to dental education and most of the questions were regarding general use of social media. So, it would be inappropriate to extrapolate the questions of our study to social media use in dental learning. We will explore disadvantages of using social media, long-term consequences, and perceptions of e-professionalism among dental students in our further study.”

The authors present a lot of data, but address/answer few meaningful questions and it remains open what their findings mean for dental education. The use of social media by dental students is not new and not surprising. Also, the observed differences between Finland and Malaysia could be more regional and be related to different social media platforms used and being popular in different global regions. Unless these two countries represent larger areas, very few people will care about country-specific findings or a comparison between two specific countries.

We agree that this study is clearly of importance in our local settings within Malaysia and Finland. We think that the findings could be generalizable beyond the immediate context. These include medical and dental schools which want to take advantage of previous documentation on the social media habits of current medical and dental undergraduates. Our findings reflect the global popularity of same platforms in contemporary society and show that most students are now heavy users of these platforms in the two culturally different countries. We have added following in the discussion:

“Another limitation of this study should be noted. It was performed in local settings in Malaysia and Finland. Each regional setting is unique. Despite its limited scope, our findings might be helpful in considering use of social media in different forms, and especially when considering students' attitudes, preferences and experiences with various platforms.”

The discussion reads more like a general review and not like a careful discussion of the findings. A discussion should also not be a reiteration of the findings, but rather address what the findings mean in the context of the wider field (dental education) and compare them to the published findings of other researchers. In summary, the discussion gets lost in small details and lacks a wider vision.

We thank the reviewer for the constructive comment. Our questions were not specific to dental education and most of the questions were regarding general use. We have now amended the format and content of the discussion section.

Even though the surveys were conducted during the COVID pandemic, there is no comparison to pre-pandemic social media use. Therefore, we really do not know whether and how the pandemic changed students’ social media use. Shorten all references to the COVID pandemic.

We appreciate this comment. We have now removed the reference to COVID pandemic from the abstract. We have also revised the text on the COVID in the discussion and added this perspective in the limitations paragraph.

There are a few interesting questions the authors should be able to address with their dataset. For example, are the social media platforms students use for their personal use correlated or the same as the platforms they use for educational purposes? Also, is their feeling of competence correlated to the use of specific platforms for dental education? Answering these questions would elevate the manuscript considerably. Also consider carefully, what of your data are really of interest to potential readers. Do not overload the manuscript with a lot of numbers, most having very little relevance. Less is often more. It might help to formulate a clear working hypothesis at the end of the introduction. What did you set out to learn, which questions did you want to answer with your research? The current purpose of the study would not encourage me to read the entire paper.

We agree and are grateful to the reviewer for the helpful suggestions. We report now new analyses between the general social media use, the self-reported competence, and the frequency of social media use for dental education. We present the results in a graph (Figure 4) to improve the ease and speed with which the findings can be located and understood. Findings are now described in chapter 3.5 as follows:

“We also analysed whether the social media platforms students use for their personal use were the same as the platforms they used for educational purposes and was their feeling of competence correlated to the use of specific platforms for dental education. Figure 4 shows a strong association between personal and educational use of platforms in both countries. In addition, knowledge of how to use the applications was associated with their use in educational purposes.”

Please clarify whether social media were officially used during any courses at any of the four schools sampled in this study

We have now added the following text in the discussion:

“In Finland, all study programs in the participating faculties do require use of social media in different forms now. It is an integral part of modern pedagogy to use social media along the other methods. The dental schools of this study in Malaysia (Manipal Melaka Medical College and University of Malaya) and Finland (University of Helsinki and University of Oulu) follow the official guidelines of social media and learning these is also an elementary study content of every dental student.”

Be careful when stating the “social media are useful for education”. The evidence is contradictory as is later mentioned in your manuscript. Also, you do NOT investigate the efficacy of social media in your project. It would be nice to mention that social media can also be a distraction, especially when used for non-educational purposes

We agree that we did not study the efficacy of social media in dental education. We have removed this sentence from the abstract. In addition, we note now in the conclusions that “Extensive use of social media can also be a distraction, especially when used for non-educational purposes.”

A paragraph outlining the limitations of the study should be added at the end of the discussion, preceding the conclusions

We have added the following paragraph in the Discussion section:

“The present study used only self-completed questionnaire and may suffer from the disadvantages of a cross-sectional online survey. The questionnaire failed to explain the underlying reasons for using specific platforms. In addition, the cross-sectional nature of our study made it impossible to assess possible rapid changes in students' use and preferences of social media. We did not study how social media communication has affected study engagement before and during the COVID-pandemic. Thus, we do not know whether and how the pandemic changed students’ social media use. Future studies will need to study trends and priorities of social media use among dental students. Another limitation of this study should be noted. It was performed in local settings in Malaysia and Finland. Each regional setting is unique. Despite its limited scope, our findings might be helpful in considering use of social media in different forms, and especially when considering students' attitudes, preferences, and experiences with various platforms.”

It would be helpful to add effect sizes to the statistical analyses.

For the analysis of the prevalence of social media use, we have reported frequency and percentage distributions. We believe that these tables and graphs best describe the basic data and allow the reader to interpret the results. In this way, the communication of the news does not rely only on the estimated coefficients which are sometimes difficult to interpret. In addition, our preliminary analyses using effect sizes for ordinal variables (e.g., Cramer’s V and Kendall’s Tau-b) revealed that these statistics do not identify associations when there is no variation in the values of the variables under study. In our data, such variables are the use of the most popular platforms Whatsapp, Youtube and Instagram. For example, all most all students used frequently or vey frequently WhatsApp in both countries.   

We agree that reporting effect sizes is useful when analysing associations between explanatory and outcome variables We have carried out new analyses on associations between the general use of social media, competence in using social media platforms and their educational use. We illustrate the findings graphically and report also Sperman’s rho correlation statistic as an effect size indicator.

Chi-square test is not the most appropriate statistical test for small response sizes. A Fisher’s exact test is recommended for such outcomes. Please consult a statistics expert.

We agree that it is preferable to calculate a significance level based on the exact distribution of the test statistic. We have not used asymptotic Chi-square test which is based on approximation when evaluating statistical significances. All statistical analyses were performed using IBM SPSS Statistics software (version 26) which reports also exact p values for Person’s Chi-square test statistic. Thus, we have obtained accurate p-values without relying on assumptions that are not met by our data. We note now in the manuscript that statistical significances “were evaluated using chi-square test with exact p-values”.

Round 2

Reviewer 2 Report

I consider the current version of the manuscript is improved and can be published as it is. 

Author Response

I consider the current version of the manuscript is improved and can be published as it is.

Thank you.

Reviewer 3 Report

The new version of this manuscript is much improved, and this reviewer appreciates the responses by the author to his/her criticism.

Especially, the discussion, which was a weak point in the previous version, is in much better shape and less a rehash of the results and more discussing the results in the context of the current knowledge.

A few smaller points which should be corrected:

(91.1% in Malaysia and 96.1% in 35 Finland used it very frequently) and Instagram (74.3% in Malaysia and 70.0% in Finland used it very frequently).

There is a word missing. Who “used it very frequently”?

“where The Finns” the article should not be capitalized: “where the Finns”

Author Response

The new version of this manuscript is much improved, and this reviewer appreciates the responses by the author to his/her criticism.

Thank you.

Especially, the discussion, which was a weak point in the previous version, is in much better shape and less a rehash of the results and more discussing the results in the context of the current knowledge.

Thank you.

A few smaller points which should be corrected:

(91.1% in Malaysia and 96.1% in 35 Finland used it very frequently) and Instagram (74.3% in Malaysia and 70.0% in Finland used it very frequently).

There is a word missing. Who “used it very frequently”?

We have corrected this to read: “(91.1% of students in Malaysia and 96.1% in Finland used it very frequently) and Instagram (74.3% of students in Malaysia and 70.0% in Finland used it very frequently)”.

“where The Finns” the article should not be capitalized: “where the Finns”

We have corrected the typo.